# Study of Liquid-Based Cytology Using Next-Generation Sequencing as a Liquid Biopsy Application in Patients with Advanced Oncological Disease

**DOI:** 10.3390/biomedicines11061578

**Published:** 2023-05-29

**Authors:** Karla Beatríz Peña, Francesc Riu, Anna Hernandez, Carmen Guilarte, Marcos Elizalde-Horcada, David Parada

**Affiliations:** 1Molecular Pathology Unit, Department of Pathology, Hospital Universitari de Sant Joan, 43202 Reus, Tarragona, Spain; karlabeatriz.pena@salutsantjoan.cat (K.B.P.); francesc.riu@salutsantjoan.cat (F.R.); anna.hernandez@salutsantjoan.cat (A.H.);; 2Institut d’Investigació Sanitària Pere Virgili, 43202 Reus, Tarragona, Spain; 3Facultat de Medicina i Ciències de la Salut, Universitat Rovira i Virgili, 43002 Reus, Tarragona, Spain

**Keywords:** liquid-based, cytology, mutation, actionable, cancer, precision, medicine

## Abstract

In patients with advanced cancer, it is necessary to detect driver mutations and genetic arrangements. If a mutation is found, targeted therapy may become an option. However, in most patients with advanced cancer, obtaining material can be challenging, and these determinations must be made based on small biopsies or cytologic samples. We analyzed the ability of liquid-based cytology to determine the mutational status in patients with advanced cancer by next-generation sequencing. We studied cytologic samples from 28 patients between 1 January 2018 and 31 December 2022. All samples were processed by next-generation sequencing using the Oncomine^®^ Precision and Comprehensive Assay Panels for Solid Tumors. Eleven male and 17 female patients with a median age of 63.75 years were included. Clinical stage IV was predominant in 21 patients. Eleven patients died, and 17 survived. The DNA and RNA concentrations were 10.53 ng/µL and 13 ng/µL, respectively. Eleven patients showed actionable mutations, and 17 showed other genomic alterations. Liquid-based cytology can be used as a component of liquid biopsy, as it allows the identification of actionable mutations in patients with advanced oncological disease. Our findings expand the utility of liquid biopsy from different body fluids or cell aspirates.

## 1. Introduction

Liquid biopsy is a technology that has evolved and improved significantly as a prognostic and predictive tool in recent years and has been considered in certain clinical situations as a complement to tissue biopsy based on the fact that it allows evaluation of the genetic panorama of solid tumors and is minimally invasive [1,2,3,4,5]. To date, liquid biopsy has proven to be useful for revealing new targets for therapy selection, tracking cancer progression during the administration of targeted therapies and immunotherapies, evaluating minimal residual disease (MRD), discovering intratumoral heterogeneity, and potentially helping with cancer diagnosis, mainly through studying tumor mutations through circulating tumor DNA [5,6]. For example, in lung cancer, ctDNA enables the detection of mutations that are potential therapeutic targets, such as epidermal growth factor receptor (*EGFR*) mutations. In addition to ctDNA, there are other elements, cellular and noncellular, that can be evaluated by liquid biopsies, such as circulating tumor cells (isolated or in groups), immune cells, circulating endothelial cells, cancer-associated fibroblasts, tumor-educated platelets, RNA, protein, and extracellular vesicles [3].

Currently, the conceptual bases of liquid biopsy can be applied to other body fluids, including urine, cerebrospinal fluid (CSF), bone marrow, saliva, or sputum, among others [3]. Thus, in the follow-up of patients with high-grade non-muscle-invasive urothelial carcinoma or to differentiate between atypical urothelial cells, urine DNA methylation may be useful [7,8,9]. Likewise, important advances are being made in the tumor pathology of the central nervous system (CNS) thanks to the study of cerebrospinal fluid (CSF), which can be analyzed for the detection, prognosis, and monitoring of cancer treatment quickly, accurately, and in real time, especially for these tumors. This CSF analysis can also uncover tumor heterogeneity and will probably replace tissue biopsy in the future. In the case of CNS tumors, the key components of liquid biopsy by means of cerebrospinal fluid mainly include circulating tumor cells (CTCs), circulating tumor nucleic acids (ctDNA, miRNA), and exosomes [10].

This present study was conducted with the aim of evaluating the utility of liquid-based cytology in patients with advanced oncological diseases, taking into account the ability of liquid-based cytology to provide real-time information on mutational status through next-generation sequencing (NGS).

## 2. Materials and Methods

### 2.1. Study Design and Patient Cohort

This is a retrospective and descriptive cohort study conducted on patients under oncological follow-up at the Medical Oncology Service of the South Catalonia Oncology Institute (Hospital Universitari de Sant Joan, Reus, Spain) between 1 January 2018 and 31 December 2022. We studied cytologic samples that 28 patients had submitted to the molecular pathology unit of our pathology department. For this study, patients with advanced oncological disease who underwent a cytological study were included. In addition, due to oncological medical conditions, those cases with the impossibility of obtaining a tissue biopsy were included. Patients in whom the cytological study was not apt for study by NGS were excluded. The patients’ clinical data were extracted from medical records, and this study was approved by the Institutional Review Board (IISPV-CEIM).

### 2.2. Liquid-Based Cytology

All cases were processed with liquid-based cytology using the Thin Prep 5000^TM^ method (Hologic Co., Marlborough, MA, USA). All cytologic material was fixed with the hemolytic and preservative solution CytoLyt^TM^ (Hologic Co., Marlborough, MA, USA). The cytologic material was spun at 3000 rpm for 5 min. The sediment was then transferred to 20 mL of PreservCyt solution (Hologic Co., Marlborough, MA, USA), kept for 15 min at room temperature, and processed with a T5000 automated processor in accordance with the manufacturer’s recommendations. Slides were obtained for each sample, fixed in 95% ethanol, and stained with Papanicolaou. In the liquid cytology study, the percentage of tumor cells present was semi quantified. Once analyzed for cellularity, an attempt was made to obtain a cellular pellet from the material in the CytoLyt^TM^ (Hologic Co., Marlborough, MA, USA). In the cases in which no more cells were obtained from the pellet, scraping of the material on the slide was carried out prior to disassembling the coverslip with xylol and hydration with alcohols at decreasing concentrations.

### 2.3. Liquid-Based Cytology Processing for Next-Generation Sequencing

Liquid-based cytology was followed by next-generation sequencing using the Oncomine^®^ Precision and Comprehensive Assay Panels for Solid Tumors (Thermo Fisher Scientific, Waltham, MA, USA) by the Ion Torrent^®^ Genexus^®^ System (Thermo Fisher Scientific, Waltham, MA, USA). Cells from the liquid-based cytology samples were obtained from pellet cells and cellular scraping, according to the following protocols: to scrape cells, the coverslip was removed, placing the slide in acetone until it came off. Next, the slide was successively placed in an acetone/xylene mix (1:1) for 5 min, in xylene for 5 min, and in 100% ethanol for 5 min, then allowed to dry at room temperature. The tumor cells were scraped and placed in a 1.5 mL tube with a digestion buffer. Pelleted cells were obtained from the liquid-based cytology samples in a sterile 50 mL tube and centrifuged for 5 min at 3000 rpm. The supernatant was discarded, and 1 mL of absolute alcohol was added to the pellet. Then, the mixture was placed in a 1.5 mL tube, and a new centrifugation was performed to remove the supernatant. Finally, the pellet was dried to room temperature until the solvent had evaporated before adding the digestion buffer. Dual DNA and RNA isolation was performed using the MagMax™ FFPE DNA/RNA Ultra Kit (Applied Biosystems ^TM^) (Thermo Fisher Scientific, Waltham, MA, USA). DNA and RNA concentrations were determined by fluorometric quantitation using Qubit Flex with Qubit 1x ds DNA HS Assay Kit and Qubit RNA HS Assay Kit (Invitrogen^TM^) (Thermo Fisher Scientific, Waltham, MA, USA) as appropriate. According to the recommendations for cytological samples of the Oncomine^®^ Precision Assay Panel for Solid Tumors, the optimal input amount of DNA was 0.67 ng/μL. Liquid-based cytology DNA aliquots (1.1 mL) were prepared. The remaining liquid-based cytology samples were stored at −80 °C as a backup. The DNA panel Oncomine^®^ Comprehensive Assay Panel for Solid Tumors was used to identify hotspot mutations (87), copy number variants (CNVs) (42), and fusion variants (51) and can also identify full-length genes (47) (Appendix A). The DNA panel Oncomine^®^ Precision Assay Panel for Solid Tumors was used to identify hotspot mutations (45), copy number variants (CNVs) (14), and fusion variants (19) (Appendix A). Ion reporter 5.18 software was used for NGS analysis with the following quality control (QC) thresholds: for molecular variation of a single nucleotide (SNV/Indel), the coverage must be at least 2 with a minimum detection cutoff frequency of 0.035% and 0.05%. To make a copy number variation (CNV) call, the following criteria must be met: mean absolute pairwise difference (MAPD) < 0.5, *p*-value < 10^−5^, CNV ratio for a copy number gain must be >1.15, and CNV ratio for one copy number loss should be <0.85.

### 2.4. Statistical Analysis

Factor analysis of mixed data (FAMD) was performed as a principal component method that makes it possible to analyze the similarity between the patients, taking into account both quantitative and qualitative variables. In this case, the two principal components explain approximately 34% of the variance in the data [11].

## 3. Results

### 3.1. Clinical Findings

Our analysis included 11 (30.29%) male and 17 (60.71%) female patients with a median age of 63.75 years (range, 39–83 years). A total of 22 (78.57%) patients had lung carcinoma, and 6 (21.43%) had breast carcinoma. Clinical stage IV was predominant in 21 (75%) patients, followed by stage III in 7 (25%) patients (Table 1). The mean follow-up was 2.5 years (range: <1 month–14 years); 7 (25%) patients had a follow-up of less than 1 year, 14 (50%) patients had a follow-up between 1 and 3 years, and 7 (25%) had a follow-up of more than 3 years. A total of 11 (39.29%) patients died from lung cancer, and the remaining 17 (60.71%) patients survived (Figure 1).

### 3.2. Cytological Findings

Cytological study was performed from lymph nodes in 13 out of 28 patients and from pleural effusion material in 9 patients. Material from bone and lung was obtained in 3 patients. A total of 63 liquid cytologies were analyzed. The cytopathology varied in each sample evaluated. Thus, the atypical population cells were arranged in small or large, rounded, three-dimensional clusters with smooth contours, papillary clusters, glandular acini, and single cells. Cytologic characteristics included an increased nuclear-to-cytoplasmic ratio and enlarged, irregular nuclei with variable pleomorphism, coarse chromatin, and prominent nucleoli (Figure 2). After cytologic study, semiquantitative quantification of cellularity was performed from each cytologic smear. Semiquantitative quantification showed a mean of 48 cells/liquid cytology (range, 10–80 cells/liquid cytology).

### 3.3. Next-Generation Sequence Findings (NGS)

After cytologic study and according to the cellularity present, in 19 out of 28 patients, DNA was extracted from cells scraped from the cytological preparation, and the remaining 9 patients’ DNA was obtained from pelleted cells. The mean DNA concentration was 10.53 ng/µL (range, 0.96–46.7 ng/µL), and the mean RNA concentration was 13 ng/µL (mean, 0.95–62.9 ng/µL). Eleven (39.29%) patients out of 28 showed actionable variants, 17 (60.71%) patients showed other genomic alterations, and 4 (14.29%) patients had no genomic alterations. Only 2 patients showed unknown significant variants.

In patients with breast cancer, 3 (50%) phosphatidylinositol-3-kinase (PI3K) (*PIK3CA*) actionable mutations were found, and *ESR1* actionable mutations were present in 1 (16.67%) patient. The *PIK3CA* (E545K) allele frequency (AF) was 3.74%, 5.40%, and 61.74%. For *ESR1* (DS38G), AF was 47.71%. In cases with *PIK3CA* mutations, the treatment option was alpelisib plus hormone therapy (Tier IA). In *BRCA1* c.5152 + 1G > A (AF: 3.75%) and *BRCA2* R3052W (AF: 6.10%), *BRCA2* c.9502-1G > A (AF: 4.27%), and *BRCA2* c.794-1G > A (AF: 3.82%) mutations, treatment options (Tier IA) were olaparib, talazoparib, bevacizumab plus olaparib, and rucaparib.

In patients with lung carcinoma, the *KRAS* G12C actionable mutation (AF: 24.55%, 51.46%, 21.32%, and 27.18%) was present in 4 (18.18%) patients, and the treatment options were sotorasib and bevacizumab plus chemotherapy (Tier IA). Additional actionable mutations, such as *BRAF* mutation (AF: 7.94%, Tier IIC (treatment options bevacizumab plus chemotherapy and ipilimumab plus nivolumab)), *EGFR* L858R mutation (AF: 59.07% and treatment options were afatinib, bevacizumab plus erlotinib, and dacomitinib (Tier IA), *RET* (*CCDC6-RET* chr10:61665880-chr10:43612032, confirmed by in situ hybridization), and *ALK* fusion, were found in 1 patient. In cases with *RET* fusion, we identified by NGS from paraffin-embedded tumors, and we confirmed the same *RET* fusion in the primary tumor diagnosis in February 2014. Table 2 summarizes the next-generation sequencing results from liquid-based cytology.

## 4. Discussion

The present study demonstrated that scraped and cellular pellet specimens obtained from liquid-based cytology can serve as a reliable source for rapid genomic analysis in patients with advanced-stage cancer since DNA was obtained to investigate mutations by means of NGS. Our results coincide with those of other studies in which the usefulness of residual liquid-based cytology specimens was assessed for NGS studies [12,13]. However, the number of tumor cells and the DNA input from liquid-based cytology seem to be the most relevant factors for an adequate NGS analysis [14]. In our study, the number of tumor cells varied in each specimen, however, this value did not limit the amount of DNA extracted to carry out the NGS study. Of the 28 samples analyzed, 26 (92.86%) had genomic mutations. Due to the absence of unsatisfactory samples, despite the low presence of tumor cells, we can assume that the quality and quantity of DNA from all samples were satisfactory for NGS.

In breast cancer, several genes have been shown to confer a clonal selective advantage to cancer cells and are involved in oncogenesis (driver mutations). *PIK3CA* is one of these genes, with a prevalence of up to 40% in primary breast cancer [14], and 75% of *PIK3CA* mutations correspond to H1047R (35%), E545K (17%), E542K (11%), N345K (6%), and H1047L (4%) [14]. In our study, the presence of *PIK3CA* was demonstrated in 3 (50%) patients, and the present mutations were E542K in 2 patients and E545K in 1, which is consistent with previous studies [14,15]. Recently, *PIK3CA* mutations have reached level 1 evidence to predict the benefit of alpelisib, an alpha-specific inhibitor of PI3K, in combination with fulvestrant in patients with advanced hormone receptor-positive/HER2-negative breast cancer previously treated with endocrine therapy, highlighting the usefulness of determining these affectations in oncological practice [16,17,18,19]. Additionally, in the context of hormone receptor-positive metastatic breast cancer, interactions in the estrogen receptor 1 (*ESR1*) gene are a frequent cause of acquired resistance to estrogen deprivation due to aromatase inhibition [20]. The prevalence of *ESR1* in primary mammary tumors is approximately 3%, however, in the metastatic setting, it is 13.6% [20]. This *ESR1* difference between primary and metastatic tumors suggests that ESR1 mutation emerges during metastasis [20]. The presence of mutated *ESR1* found in a patient with pleural effusion is consistent with what has been described in metastatic breast cancers. In addition, the demonstration of *ESR1* mutations has therapeutic implications, as these patients may benefit from biological targets such as mTOR and cyclin-dependent kinases (CDKs) 4/6 [20]. Finally, in 2 patients with breast cancer, no actionable variants were shown; however, the presence of mutations in *NOTCH1* and *TP53* was demonstrated, which can be used as follow-up biomarkers in these patients.

In our study, of the 22 patients with metastatic non-small cell lung carcinoma (NSCLC) studied by NGS, 17 showed evidence of different gene alterations in actionable and nonactionable variants. The *KRAS-G12C* mutation was predominant in 4 patients, and mutations in G12D, G12A, and G12F were also present. This mutational variability in KRAS has been described in other investigations, and these mutational subtypes are related to the cooccurrence of alterations in pathways associated with cancer, such as *TP53*, *STK11*, and *KEAP1* [21]. Our work confirms these findings. Currently, approximately 80% of patients with lung cancer are diagnosed in the advanced stages of the disease [21], which raises the need to identify target genomic alterations as a standard of care to guide the selection of optimal therapy. In this regard, the presence of *KRAS*-mutated NSCLC may be more sensitive to new immunotherapy agents. Indeed, in the CheckMate 057 (Checkpoint Pathway and Nivolumab Clinical Trial Evaluation 057) trial, a subgroup of patients with NSCLC harboring KRAS mutations experienced greater benefit from the anti-ePD-1 checkpoint inhibitor nivolumab [21,22,23,24,25,26]. Another mutational alteration found in our series was the *EGFR-L858R* mutation in a patient with pleural effusion. The implication of this finding is that patients with lung adenocarcinoma harboring *EGFR* mutations can be treated with *EGFR* tyrosine kinase inhibitors (TKIs), improving survival in such patients [27]. In addition, the presence of mutated *EGFR-L858R* is related to the invasive capacity of lung adenocarcinomas that present this mutation and is associated with malignant pleural effusion, as demonstrated in our patient [28,29].

Our study has some limitations. (a) The sample includes two types of solid cancers, breast and lung, with material obtained from different locations. This raises the possibility that differences in cell turnover between different tumor types, their histopathological characteristics, and their biological behavior confer differences in the sensitivity of liquid-based cytology. (b) The cellular concentration and therefore that of the DNA can vary according to the tumor biorhythm and the tumor–tumor environment interaction; thus, the analyzed DNA samples can present locoregional variations and show different cell clones. (c) Because the samples were taken at advanced stages, data on early mutational alterations could not be obtained. (d) Finally, the number of cases is not high enough, however, we were able to demonstrate the utility of liquid cytology for the determination of gene alterations by means of next-generation sequencing.

## 5. Conclusions

Liquid-based cytology can be used as a component of liquid biopsy, as it allows the identification of actionable mutations in patients with advanced oncological disease. In addition, the presence of other genomic alterations can be used to assess the response to treatment and in the follow-up of these patients. Our findings expand the utility of liquid biopsy from different body fluids or cell aspirates.

## Figures and Tables

**Figure 1 biomedicines-11-01578-f001:**
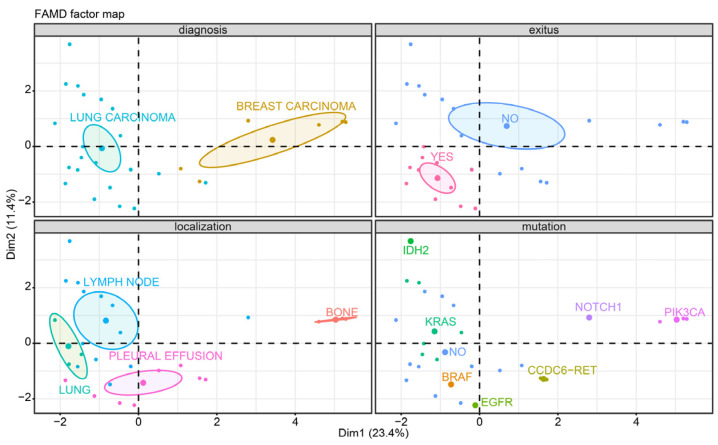
Factor analysis of mixed data. Lung adenocarcinoma was the predominant advanced cancer diagnosis. Most patients were exitus, and variable locations are shown. Mutational status in lung carcinoma was heterogeneous.

**Figure 2 biomedicines-11-01578-f002:**
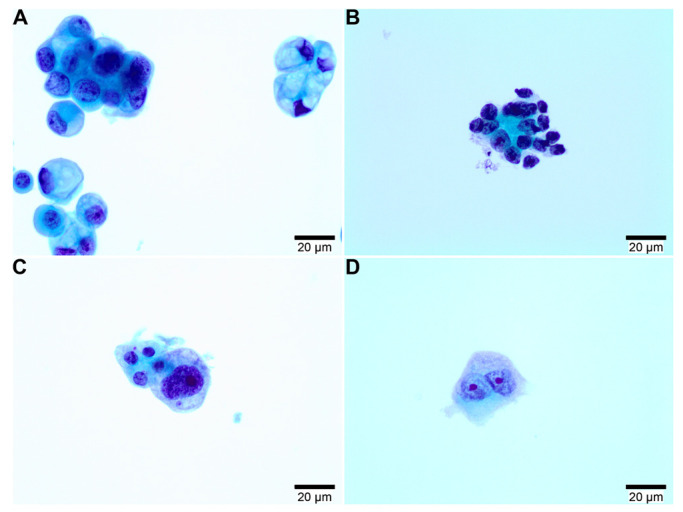
Cytological findings from liquid-based cytology. (**A**,**B**) Characteristic finding of atypical cells. Small, rounded, three-dimensional clusters are shown, and hyperchromatic nuclei and cytoplasm vacuolization are seen. (**C**,**D**) Isolated cell group with evidence of cytologic atypia. Some cytoplasm has a vacuolated aspect with hyperchromatic nuclei (**C**) and two isolated atypical cells with marked cytologic atypia, consistent with carcinoma (**D**) (Papanicolaou staining. Direct magnification 40×).

**Table 1 biomedicines-11-01578-t001:** Clinical characteristics of the patients (N = 28).

	Total (N = 28 Patients)
Age, median (range)	63.75 (39–83)
Sex, n (%)	
Male	11 (39.29%)
Female	17 (60.71%)
Stage, n (%)	
III	7 (25%)
IV	21 (75%)
Tumor location, n (%)	
Lymph node	13 (46.3%)
Pleural effusion	9 (32.5%)
Bone	3 (10.71%)
Lung	3 (10.71%)
Primary tumor, n (%)	
Lung	22 (78.57%)
Breast	6 (21.43%)
Follow-up time (years)	
<1	7 (25%)
1–3	14 (50%)
>3–5	3 (10.71%)
>5	4 (14.29%)
Evolution	
Died	11 (39.9%)
Survived	17 (60.71%)

**Table 2 biomedicines-11-01578-t002:** Next-generation sequencing findings summary from liquid-based cytology (N = 28).

Patient	Primary Tumor	Cytological Location	Actionable Gen	Other Alterations	Unknown Significant Variants
1	BREAST	BONE	*PIK3CA*, *BRCA1*, *BRCA2*	*ATM*, *PDGFRA*, *ERBB2*, *PTEN*, *KRAS*, *NF1*, *NBN*, *MRE11*, *POLE*, *ATR*, *FANCI*, *TP53*, *ARID1A*, *SMARCA4*, *FLT3*, *CREBBP*, *RB1*, *CBL*	*NONE*
2	BREAST	BONE	*PIK3CA*	*TP53*, *MDM4*, *MYC*, *ARID*, *POLE*, *FANCA*, *ERBB2*, *NONE*, *NOTCH3*, *FGFR1*, *NOTCH2*	*NONE*
3	BREAST	PLEURAL EFFUSION	*ESR1*	*NONE*	*NONE*
4	BREAST	BONE	*PIK3CA*	*NONE*	*NONE*
5	LUNG	LYMPH NODE	*KRAS G12C*	*NONE*	*NONE*
6	LUNG	LYMPH NODE	*NONE*	*KRAS G12A*	*NONE*
7	LUNG	LYMPH NODE	*NONE*	*NONE*	*NONE*
8	LUNG	LYMPH NODE	*KRAS G12C*	*ERBB3, TP53*	*NONE*
9	BREAST	LYMPH NODE	*NONE*	*NOTCH1*	*NONE*
10	LUNG	LUNG	*NONE*	*NONE*	*NONE*
11	LUNG	LYMPH NODE	*BRAF*	*NONE*	*NONE*
12	LUNG	PLEURAL EFFUSION	*EGFR*	*TP53*	*NONE*
13	LUNG	PLEURAL EFFUSION	*NONE*	*KRAS G12D*	*NONE*
14	LUNG	LUNG	*NONE*	*IDH2*	*NONE*
15	LUNG	LYMPH NODE	*NONE*	*KRAS G12F*, *TP53*	*NONE*
16	LUNG	PLEURAL EFFUSION	*NONE*	*KRAS G12D*	*NONE*
17	LUNG	LYMPH NODE	*KRAS G12C*	*NONE*	*NONE*
18	LUNG	LYMPH NODE	*NONE*	*TP53*	*NONE*
19	LUNG	LYMPH NODE	*NONE*	*TP53*	*NONE*
20	LUNG	LYMPH NODE	*NONE*	*TP53*, *AR*	*NONE*
21	LUNG	PLEURAL EFFUSION	*NONE*	*NONE*	*NONE*
22	LUNG	LYMPH NODE	*KRAS G12C*	*NONE*	*NONE*
23	LUNG	PLEURAL EFFUSION	*NONE*	*BRAF*	*NONE*
24	LUNG	LUNG	*NONE*	*NONE*	*NONE*
25	BREAST	PLEURAL EFFUSION	*NONE*	*TP53*	*PTEN (DELETION)*
26	LUNG	LYMPH NODE	*NONE*	*KRAS G12V*	*NONE*
27	LUNG	PLEURAL EFFUSION	*CCD6-RET (FUSION)*	*ALK (FUSION)*	*FGFR3*
28	LUNG	PLEURAL EFFUSION	*NONE*	*ALK (FUSION)*	*NONE*

## Data Availability

All data generated or analyzed during this study are included in this article. Further inquiries can be directed to the corresponding author.

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
