# Peer review of "Study of Liquid-Based Cytology Using Next-Generation Sequencing as a Liquid Biopsy Application in Patients with Advanced Oncological Disease"

_biomedicines, 2023, doi:10.3390/biomedicines11061578_

Round 1
Reviewer 1 Report
In recent years, the rapid development of next-generation sequencing (NGS) technologies has led to a significant reduction in sequencing cost with improved accuracy. Liquid-based cytology can be used as a component of liquid biopsy.This paper focuses on the application of liquid-based cytology to different body fluids or cell aspirates in patients with advanced oncological diseases, taking into account the ability of liquid-based cytology to provide real-time information on mutational status through next-generation sequencing .
References should be enriched.
Chen, M., Zhao, H. Next-generation sequencing in liquid biopsy: cancer screening and early detection. Hum Genomics 13, 34 (2019). https://doi.org/10.1186/s40246-019-0220-8

Author Response
Reviewer 1:
In recent years, the rapid development of next-generation sequencing (NGS) technologies has led to a significant reduction in sequencing cost with improved accuracy. Liquid-based cytology can be used as a component of liquid biopsy. This paper focuses on the application of liquid-based cytology to different body fluids or cell aspirates in patients with advanced oncological diseases, taking into account the ability of liquid-based cytology to provide real-time information on mutational status through next-generation sequencing .
1) Point 1: References should be enriched.
Chen, M., Zhao, H. Next-generation sequencing in liquid biopsy: cancer screening and early detection. Hum Genomics 13, 34 (2019). https://doi.org/10.1186/s40246-019-0220-8
Response to reviewer 1:
Dear reviewer, thank you for your valuable comment and time. We have enriched the study by placing the suggested reference:
1.1) Response to point 1: We have enriched our paper placing the suggested reference:
- Chen, M., Zhao, H. Next-generation sequencing in liquid biopsy: cancer screening and early detection. Hum Genomics 2019, 13, 34.
Reviewer 2 Report
I have reviewed a manuscript titled “Study of Liquid-Based Cytology using Next Generation Sequencing as a Liquid Biopsy Application in Patients with Advanced Oncological Disease”. There are some concerns that the authors would like to further discuss or address. These concerns are as follows:
1. Maybe I have overlooked, the FDR, read length (100 or 150bp?), read depth, read nature like paired-end or singled-end, what library prep used and what machine was run (Hi-Seq, Mid-Seq, NovaSeq or what) were not found in the manuscript, nor more basic information about the NGS like is it Whole exome sequencing or whole genome sequencing or what kind of sequencing is it? was not mentioned. I just know there are some data and that is.
2. Maybe I have overlooked again, it seems that the inclusion and exclusion criteria was not well known in the method. I know it is a retrospective study, but still some kind of inclusion or exclusion criteria seems necessary? correct me if I am wrong.
3. Further, it seems that the population size is pretty small. Why only these 28 subjects were included? Is that these 28 subjects can represent the whole population of the region under study? If not, what are the significant of these 28 subjects?
4. It was mentioned that 26 out of 28 subjects got genetic mutations, while from Table 2, Maybe I have overlooked or got it wrong, it seems that 4 subjects (i.e. 7, 10, 21 and 24) have no actionable mutations nor other alternations nor VUS. Can the authors please clarify or explain on that? And if Table 2 is correct, is seems that there are 11 out of 28 subjects got no actionable variants found, which is less than half of the sample size, which is not a quite nice results on that, how the authors account for that?
5.a. The genetic analysis is pretty much superficial and suboptimal in terms of scientific. Not even the genetic change of the variants or genomic position were mentioned (e.g. BRAF:c.1799T>G) (the variant change mentioned is amino acid changes, which might not infer back to the genetic changes due to the wobbling of codons), not to mention the proper nomenclature of these variants according to ACMG guideline. Next, what further information or analysis can be obtained from these genetic results from NGS? Are these mutations located at the hotspot? Are these compound heterozygous mutations in these subjects? It was reported that there were only 2 subjects got “no genomic mutations”, probably the variants are novel (at least not common mutations, or the read depth of the sequencing is too low for concrete results) is there any possibility to further work on that such as, whole exome or whole genome sequencing to find out the genetic causes? These will be more interesting (in terms of scientific research) than the commonly variants.
5.b. While in terms of clinical, the results presented were also suboptimal. it seems that the change in clinical management on the subjects due to the genetics findings was not mentioned in these subjects. In the manuscript, although there were a lot of treatment options presented, it does not necessarily mean that were the treatment received by the subjects. Not to mentioned are these treatment plan changes made because of the genetic findings as the physician can already prescribed those treatments simply based on clinical findings before any genetic cause detection by NGS.
6. The novelty of the main aim of this study is not clearly known. This study is using sequencing kits commercially available for liquid biopsies. The validation (scientifically and clinically) of these methods has already been tested by the manufacturers (and probably approved by FDA). While it is totally fine to use commercially available kits for research, but it seems that there were no any innovative modifications nor analysis on any of the methods or results. Thus, I can hardly see any novelty in this study. I do not understand why the authors need to re-validate or show it again with these 28 subjects. Maybe the authors can discuss the novelty and scientific significance on this study.
7. It was mentioned that the study was approved by the institutional review board, but just to confirm that whether the subjects is necessary to have written consent to participate in the study as well as to be published.
8. There are a lot of portions in the method (line 106 to line 141) mentioning the gene mutations hotspot can be detected by the commercial kits. It is great to know that but is it worthy to spend the precious manuscript main text area for that? Could it be in the supplementary file only?
9. Also, it was mentioned that the optimal amount of DNA to be loaded is 0.67ng/ul (or other applicable values), however, maybe I have overlooked again, it seems that there was no other information on the actual amount loaded. I guess the optimal and actual sample loaded might be different, can the authors clarify?
10. The discussion is basically a clinical literature review of the variants found in the subjects. Apart from the variant itself, I do not see any relationship to these 28 subjects.
11. The authors mentioned that the genome analysis is rapid and reliable, while the turn-around-time is not known in the manuscript.
12. It seems that the Figure 1 is way too small and resolution is too low to read the words inside. Is it possible to improve that?
Author Response
Reviewer 2:
Dear reviewer, thank you for your valuable comment and time. We have collected your comments in order to improve our work and so that it can be assessed for possible publication. Here are the changes made:
I have reviewed a manuscript titled “Study of Liquid-Based Cytology using Next Generation Sequencing as a Liquid Biopsy Application in Patients with Advanced Oncological Disease”. There are some concerns that the authors would like to further discuss or address. These concerns are as follows:
1) Point 1. Maybe I have overlooked, the FDR, read length (100 or 150bp?), read depth, read nature like paired-end or singled-end, what library prep used and what machine was run (Hi-Seq, Mid-Seq, NovaSeq or what) were not found in the manuscript, nor more basic information about the NGS like is it Whole exome sequencing or whole genome sequencing or what kind of sequencing is it? was not mentioned. I just know there are some data and that is.
1.1) Response to point 1: Mean read lengths for these assays range between 56-101bp for DNA and 40-96bp for RNA. Effective read depth was set on 500. Analyses were performed with the Genexus™ Integrated Sequencer, which performs library preparation, sequencing, analysis, and reporting automatically. The assayed panel were Oncomine Comprehensive Assay and Oncomine Precision Assay, with Ion AmpliSeq (or AmpliSeq HD) chemistry respectively, which enables the detection of main hotspots of cancer genes.
2) Point 2. Maybe I have overlooked again, it seems that the inclusion and exclusion criteria was not well known in the method. I know it is a retrospective study, but still some kind of inclusion or exclusion criteria seems necessary? correct me if I am wrong.
2.1) Response to point 2: Thank you for your comment and following your indication we have included the following:
Line 66-70: For the study, patients with advanced oncological disease who underwent a cytological study were included. In addition, due to oncological medical conditions, those cases with impossibility of obtaining a tissue biopsy were included. Patients in whom the cytological study was not apt for study by NGS were excluded.
3) Point 3. Further, it seems that the population size is pretty small. Why only these 28 subjects were included? Is that these 28 subjects can represent the whole population of the region under study? If not, what are the significant of these 28 subjects?
3.1) Response to point 3: Thank you for your comment and we would like to highlight the following:
The sample evaluated corresponds to a specific sub-population of patients with advanced oncological disease. In these clinical conditions, obtaining material for analysis by NGS represents a challenge. Our objective was based on the utility of liquid cytology to carry out a study for NGS, which provided important information on the mutational status in a particular clinical condition. Due to this, our sample could be interpreted as limited, however, the patients studied are a representative sample in daily oncology practice, since most of these represent cases with various lines of treatment and the group involved in them requires other tools for their possible diagnosis. inclusion in very specific clinical trials, with extremely restrictive criteria.
4) Point 4. It was mentioned that 26 out of 28 subjects got genetic mutations, while from Table 2, Maybe I have overlooked or got it wrong, it seems that 4 subjects (i.e. 7, 10, 21 and 24) have no actionable mutations nor other alternations nor VUS. Can the authors please clarify or explain on that? And if Table 2 is correct, is seems that there are 11 out of 28 subjects got no actionable variants found, which is less than half of the sample size, which is not a quite nice results on that, how the authors account for that?
4.1) Response to point 4: Thank you for your comment and after the review we have corrected the total number of patients in which the NGS study did not show the presence of genomic alterations.
Line 223: 4 (14.29%) patients.
5) Point 5.a. The genetic analysis is pretty much superficial and suboptimal in terms of scientific. Not even the genetic change of the variants or genomic position were mentioned (e.g. BRAF:c.1799T>G) (the variant change mentioned is amino acid changes, which might not infer back to the genetic changes due to the wobbling of codons), not to mention the proper nomenclature of these variants according to ACMG guideline. Next, what further information or analysis can be obtained from these genetic results from NGS? Are these mutations located at the hotspot? Are these compound heterozygous mutations in these subjects? It was reported that there were only 2 subjects got “no genomic mutations”, probably the variants are novel (at least not common mutations, or the read depth of the sequencing is too low for concrete results) is there any possibility to further work on that such as, whole exome or whole genome sequencing to find out the genetic causes? These will be more interesting (in terms of scientific research) than the commonly variants.
5.1) Response to point 5.a: The genetic results of the NGS are directed above all to clinical applicability, especially to decide the administration (or not) of certain drugs with therapeutic targets. Most of all detected mutation in Oncomine kits are located at hotspots. In the case of those two subjects with “no genomic mutations” we propose to change that sentence to “no mutation identified” in order to consider possible variants not detected due to technical procedures. For the moment, we do not have the methodology to include whole exome/genome sequencing but we will consider it for future investigations.
6) Point 5.b. While in terms of clinical, the results presented were also suboptimal. it seems that the change in clinical management on the subjects due to the genetics findings was not mentioned in these subjects. In the manuscript, although there were a lot of treatment options presented, it does not necessarily mean that were the treatment received by the subjects. Not to mentioned are these treatment plan changes made because of the genetic findings as the physician can already prescribed those treatments simply based on clinical findings before any genetic cause detection by NGS.
6.1) Response to point 5.b: Thank you for your comment, and we would like to highlight the following:
In our study group, the need to carry out an NGS study was raised in the molecular study committee of our center. In this committee, the clinical condition of the patient and the need to offer a specific therapeutic option are discussed, hence the need to obtain an actionable path. In addition, our center contributes to different clinical trials, which implies the need to obtain extensive tumor information.
7) Point 6. The novelty of the main aim of this study is not clearly known. This study is using sequencing kits commercially available for liquid biopsies. The validation (scientifically and clinically) of these methods has already been tested by the manufacturers (and probably approved by FDA). While it is totally fine to use commercially available kits for research, but it seems that there were no any innovative modifications nor analysis on any of the methods or results. Thus, I can hardly see any novelty in this study. I do not understand why the authors need to re-validate or show it again with these 28 subjects. Maybe the authors can discuss the novelty and scientific significance on this study.
7.1) Response to point 6: Thank you for your comment, and our study did not focus on the validation of the NGS using commercial kits. Our study focused on the capacity of liquid cytology as a valid tool in patients with advanced disease, in whom obtaining samples represents, on certain occasions, as explained above, a challenge from the clinical point of view. In our study we did not re-validate our group of patients.
8) Point 7. It was mentioned that the study was approved by the institutional review board, but just to confirm that whether the subjects is necessary to have written consent to participate in the study as well as to be published.
8.1) Response to point 7: As you point out, the study was approved by our institutional review board and it detailed the reasons why informed consent could be waived.
9) Point 8. There are a lot of portions in the method (line 106 to line 141) mentioning the gene mutations hotspot can be detected by the commercial kits. It is great to know that but is it worthy to spend the precious manuscript main text area for that? Could it be in the supplementary file only?
9.1) Response to point 8: Thank you for your comment, and following your indication we have included this section in supplementary material (S1 and S2).
10) Point 9. Also, it was mentioned that the optimal amount of DNA to be loaded is 0.67ng/ul (or other applicable values), however, maybe I have overlooked again, it seems that there was no other information on the actual amount loaded. I guess the optimal and actual sample loaded might be different, can the authors clarify?
10.1) Response to point 9: Dear reviewer, thank you for your interesting comment and according to our review, we did not find studies that compare the concentration of circulating nuclear acids and nuclear acids in liquid cytology. This is a very interesting topic, and it raises future perspectives on the usefulness of liquid biopsy, in addition to trying to explain questions about tumor biorhythm, the turn-over of nulceic acids, among other factors. Additionally, taking into account your question, we tried to correlate the cellularity in the liquid cytology and the concentrations of both DNA and RNA, finding no correlation between these variables. Below are the graphs (Please see PDF). Also, Genexus system makes the appropriate dilution of each sample to reach the optimal amount of DNA in all samples.
11) Point 10. The discussion is basically a clinical literature review of the variants found in the subjects. Apart from the variant itself, I do not see any relationship to these 28 subjects.
11.1) Response to point 10: Thank you for your comment and we would like to point out the following:
Our discussion focused on different molecular aspects evaluated in our series, with predictive and prognostic implications. We were even able to demonstrate the pathophysiological and molecular correlation with the presence of pleural effusion due to lung adenocarcinoma corroborated by a specific mutational state.
12) Point 11. The authors mentioned that the genome analysis is rapid and reliable, while the turn-around-time is not known in the manuscript.
12.1) Response to point 11: The normal mean time of NGS results in our service is: 72 hours (OPA), and 5 days (OCA).
13) Point 12. It seems that the Figure 1 is way too small and resolution is too low to read the words inside. Is it possible to improve that?
13.1) Response to point 12: Following your instructions we have improved the figures.

Reviewer 3 Report
The study is nicely presented. Unfortunately the number of cases is not high enough. Please mention it as limitation of thia study.
It would be interesting to see is there any correlation with cell-free nucleic acids in the same liquid biopsy sample. Are there studies addressing this?
Please refer for the possible connection between cf-nucleic acids and tumor cells in liquid biopsy samples. What is the clinical importance of this work?
Please improve the quality of your figures.
Author Response
Reviewer 3:
The study is nicely presented. Unfortunately the number of cases is not high enough. Please mention it as limitation of this study.
It would be interesting to see is there any correlation with cell-free nucleic acids in the same liquid biopsy sample. Are there studies addressing this?
Please refer for the possible connection between cf-nucleic acids and tumor cells in liquid biopsy samples. What is the clinical importance of this work?
Please improve the quality of your figures.
Respones to Reviewer 3:
Dear reviewer, thank you for your valuable comment and time. We have collected your comments in order to improve our work and so that it can be assessed for possible publication. Here are the changes made:
1) Point 1. Unfortunately the number of cases is not high enough. Please mention it as limitation of this study.
1.1) Response to point 1: Thank you for your comment, and based on your suggestion we have placed the number of patients within the limitations in the discussion, as follows:
Finally, the number of cases is not high enough, however we were able to demonstrate the utility of liquid cytology for the determination of gene alterations by means of next generation sequence.
2) Pont 2. It would be interesting to see is there any correlation with cell-free nucleic acids in the same liquid biopsy sample. Are there studies addressing this?
2.1) Response to point 2: Dear reviewer, thank you for your interesting comment and according to our review, we did not find studies that compare the concentration of circulating nuclear acids and nuclear acids in liquid cytology. This is a very interesting topic, and it raises future perspectives on the usefulness of liquid biopsy, in addition to trying to explain questions about tumor biorhythm, the turn-over of nulceic acids, among other factors.
Thank you for your comment, and based on your suggestion we have placed the number of patients within the limitations in the discussion, as follows:
Additionally, taking into account your question, we tried to correlate the cellularity in the liquid cytology and the concentrations of both DNA and RNA, finding no correlation between these variables. Below are the graphs :
Please see PDF.
3) Point 3. Please refer for the possible connection between cf-nucleic acids and tumor cells in liquid biopsy samples. What is the clinical importance of this work?
3.1) Response to point 4: In reference to your question, we would like to present the following:
cfDNA refers to DNA fragments present outside of cells in body fluids such as plasma, urine, and cerebrospinal fluid (CSF). In plasma, the majority of cfDNA originates from leukocytes, and only a small fraction is tumour-derived, known as circulating tumour DNA (ctDNA). ctDNA can contain mutations missed in biopsy studies because of tumour heterogeneity or lesions in distant sites, and is generally found in larger quantities in the bloodstream than circulating tumour cells. The concentration of ctDNA varies among patients, and differs according to the type, location and stage of cancer, with some producing extremely low concentrations. The half-life of ctDNA is still unclear, although fetal cfDNA studies have highlighted its short duration (16 min to 2 h). This instability, which might be an issue at the preanalytical level, can be used to our advantage by providing a very dynamic tool for tracking treatment response within hours. Finnaly, the relationship between tumour biology and ctDNA release into the circulation is still unclear. Experimental data have shown that cfDNA is highly fragmented, in a largely chromatosomal pattern – indicating potential associations with nucleosomes and transcription factors. ctDNA fragments are slightly shorter, with the majority being <167 bp. The importance of our study lies in the utility of liquid-based cytology as a surrogate for liquid biopsy. Thus, liquid cytology allows the identification of genomic alterations that can be used as a prognostic or predictive factor in cancer patients with advanced disease.
4) Point 4. Please improve the quality of your figures.
4.1) Response to point 4: Following his instructions we have improved the figures.
Thank you very much, in advance, for your consideration.

Round 2
Reviewer 2 Report
Thank you very much for the authors response, just a brief follow-up of your response.
1. To your response 5.1 and 6.1, it was understand that not always studies will have scientific significance, but for the clinical significance, it was always great to know people using new technologies like sequencing in helping out patients diagnosis, but it was mentioned in Table 2 (if my understanding is correct) that, 11 out of 28 subjects (almost half of the sample size) got no actionable variants found. The applicability of the kit (I will not say the sequencing technologies as it is helped a lot of other patients in other parts of the world) is questionable, and thus, the clinical significance of this study is also questionable.
2. To your response 7.1, thanks for pointing out clearly the main focus of the study. To my understanding, buccal swab, (is considered as a kind of liquid-based cytology sample) is widely used for whole genome sequencing, and in cancer sequencing, even in commercial genome sequencing service. Therefore, the “capacity of liquid cytology” has already been proven. Indeed, for the sample type used in this manuscript, such as the use of Pleural Effusions for cancer sequencing, has also been reported in literatures (e.g. PMC:7712846), same for lymph node samples (e.g. PMC:8492772), not to mention other commonly known samples like bone marrow and blood. Thus, I am not quite sure whether the “capacity of liquid cytology” can still be the unique and significance of this manuscript.
To be honest, I am not trying to give the authors a hard time, but as a reviewer and also a reader of academic journals, the first question I would ask myself is what is the significance or importance of this manuscript? Maybe be the samples are very hard to get for your sequencing, but it seems that some others has already done similar things and made this study “not so special” (if not no significance). Therefore, I think the authors might need to think or write from the other angle, so that the manuscript can be considered as something novel and important as well as contributing to the science community.
Author Response
Special Issue Editor
Biomedicines
Reus, May 17, 2023
Dear Academic Editor,
Enclosed please find the manuscript entitled “Study of Liquid-Based Cytology using Next Generation Sequencing as a Liquid Biopsy Application in Patients with Advanced Oncological Disease” which we are resubmitting (Manuscript resubmission Biomedicines-2337675) as an original contribution for editorial consideration by your journal. All the authors have made a substantial contribution to the information or material submitted for publication and we have read and approved the final manuscript. None of us have any direct or indirect commercial financial incentive associated with publishing the article and we have indicated the source of extra-institutional funding, particularly that provided by commercial sources. Finally, the manuscript, or portions thereof, is not under consideration by any other journal or electronic publication, and has not been published and is not under current consideration elsewhere.
We appreciate your time in evaluating our study, as well as that of the reviewers. Below, we point out the changes made by the comments received, as follows:
Reviewer 2:
- To your response 5.1 and 6.1, it was understand that not always studies will have scientific significance, but for the clinical significance, it was always great to know people using new technologies like sequencing in helping out patients diagnosis, but it was mentioned in Table 2 (if my understanding is correct) that, 11 out of 28 subjects (almost half of the sample size) got no actionable variants found. The applicability of the kit (I will not say the sequencing technologies as it is helped a lot of other patients in other parts of the world) is questionable, and thus, the clinical significance of this study is also questionable.
- To your response 7.1, thanks for pointing out clearly the main focus of the study. To my understanding, buccal swab, (is considered as a kind of liquid-based cytology sample) is widely used for whole genome sequencing, and in cancer sequencing, even in commercial genome sequencing service. Therefore, the “capacity of liquid cytology” has already been proven. Indeed, for the sample type used in this manuscript, such as the use of Pleural Effusions for cancer sequencing, has also been reported in literatures (e.g. PMC:7712846), same for lymph node samples (e.g. PMC:8492772), not to mention other commonly known samples like bone marrow and blood. Thus, I am not quite sure whether the “capacity of liquid cytology” can still be the unique and significance of this manuscript.
To be honest, I am not trying to give the authors a hard time, but as a reviewer and also a reader of academic journals, the first question I would ask myself is what is the significance or importance of this manuscript? Maybe be the samples are very hard to get for your sequencing, but it seems that some others has already done similar things and made this study “not so special” (if not no significance). Therefore, I think the authors might need to think or write from the other angle, so that the manuscript can be considered as something novel and important as well as contributing to the science community.
Dear reviewer, thank you for your valuable comment and time. We have collected your comments in order to improve our work and so that it can be assessed for possible publication. These are the comments to their considerations:
1) Point 1. To your response 5.1 and 6.1, it was understand that not always studies will have scientific significance, but for the clinical significance, it was always great to know people using new technologies like sequencing in helping out patients diagnosis, but it was mentioned in Table 2 (if my understanding is correct) that, 11 out of 28 subjects (almost half of the sample size) got no actionable variants found. The applicability of the kit (I will not say the sequencing technologies as it is helped a lot of other patients in other parts of the world) is questionable, and thus, the clinical significance of this study is also questionable.
1.1) Response to Point 1: We agree with your point of view that clinical application is one of the main objectives of liquid biopsy. We would also like to highlight that in the continuum of the cancer patient, knowledge of actionable pathways is fundamental due to the implications that they may have as a predictive factor. However, the demonstration of non-actionable alterations may have prognostic applications and to assess the response to treatment. Lastly, the knowledge of these gene modifications should be reported as it allows increasing the knowledge of possible oncogenic pathways in patients with advanced oncological disease.
2) Point 2. To your response 7.1, thanks for pointing out clearly the main focus of the study. To my understanding, buccal swab, (is considered as a kind of liquid-based cytology sample) is widely used for whole genome sequencing, and in cancer sequencing, even in commercial genome sequencing service. Therefore, the “capacity of liquid cytology” has already been proven. Indeed, for the sample type used in this manuscript, such as the use of Pleural Effusions for cancer sequencing, has also been reported in literatures (e.g. PMC:7712846), same for lymph node samples (e.g. PMC:8492772), not to mention other commonly known samples like bone marrow and blood. Thus, I am not quite sure whether the “capacity of liquid cytology” can still be the unique and significance of this manuscript.
2.1) Response to Point 2: Thank you for your comment and once again we agree with your point of view on liquid biopsy. However, we would like to highlight that in this study, in addition to using liquid-based cytologies for genomic study, we used scraping samples from cytological slides, as indicated in the materials and methods. This obtaining of material may be key in patients in whom it is not possible to obtain new material to be able to carry out genomic determination. This methodology implies that the fixation and staining used in liquid-based cytology obtained from different locations does not affect the genomic detection capacity. Furthermore, our study shows that the material processed and archived by liquid-based cytology can be useful in clinical routine for the demonstration of genomic alterations.
3) Point 3. To be honest, I am not trying to give the authors a hard time, but as a reviewer and also a reader of academic journals, the first question I would ask myself is what is the significance or importance of this manuscript? Maybe be the samples are very hard to get for your sequencing, but it seems that some others has already done similar things and made this study “not so special” (if not no significance). Therefore, I think the authors might need to think or write from the other angle, so that the manuscript can be considered as something novel and important as well as contributing to the science community.
3.1) Response to Point 3: Dear reviewer, we would like to reiterate our gratitude for your valuable comments, there is no doubt that your point of view and comments help to improve our study and we hope that all the modifications made have improved the quality of our paper. We believe that little has been described about the different possibilities and methodologies that we can implement in laboratories that use liquid biopsy. Thus, our study provides information in this regard.
Thank you very much, in advance, for your consideration.
Sincerely,
Dr. David Parada
Corresponding author
David Parada D, MD, PhD. Unit of Molecular Pathology, Pathology Service. University Hospital of Sant Joan, Faculty of Medicine, IISPV, “Rovira i Virgili” University, Reus, Tarragona, Spain.